# The Barcelona Predictive Model of Clinically Significant Prostate Cancer

**DOI:** 10.3390/cancers14061589

**Published:** 2022-03-21

**Authors:** Juan Morote, Angel Borque-Fernando, Marina Triquell, Anna Celma, Lucas Regis, Manel Escobar, Richard Mast, Inés M. de Torres, María E. Semidey, José M. Abascal, Carles Sola, Pol Servian, Daniel Salvador, Anna Santamaría, Jacques Planas, Luis M. Esteban, Enrique Trilla

**Affiliations:** 1Department of Urology, Vall d’Hebron Hospital, 08035 Barcelona, Spain; mtriquell@vhebron.net (M.T.); acelma@vhebron.net (A.C.); lregis@vhebron.net (L.R.); jplanas@vhebron.net (J.P.); etrilla@vhebron.net (E.T.); 2Department of Surgery, Universitat Autònoma de Barcelona, 08193 Barcelona, Spain; 3Department of Urology, Hospital Universitario Miguel Servet, IIS-Aragon, 50009 Zaragoza, Spain; aborque@comz.org; 4Department of Radiology, Vall d´Hebron Hospital, 08035 Barcelona, Spain; mescobar@vhebon.net (M.E.); rmast@vhebron.net (R.M.); 5Department of Pathology, Vall d´Hebron Hospital, 08035 Barcelona, Spain; itorres@vhebron.net (I.M.d.T.); mesemidey@vehbron.net (M.E.S.); 6Department of Morphological Sciences, Universitat Autònoma de Barcelona, 08193 Barcelona, Spain; 7Department of Urology, Parc de Salut Mar, 08003 Barcelona, Spain; jmabascalj@gmail.com (J.M.A.); csolamarques@psmar.cat (C.S.); 8Department of Urology, Hospital Germans Trias I Pujol, 08916 Badalona, Spain; p.serviangermanstrias@gencat.cat (P.S.); dsalvadorhidalgo@gmail.com (D.S.); 9Urology Research Group, Vall d´ Hebron Research Institute, 08035 Barcelona, Spain; anna.santamaria@vhir.org; 10Department of Applied Mathematics, Escuela Universitaria Politécnica La Almunia, Universidad de Zaragoza, 50100 Zaragoza, Spain; lmeste@unizar.es

**Keywords:** clinically significant prostate cancer, magnetic resonance imaging, predictive model, risk calculator

## Abstract

**Simple Summary:**

Magnetic-resonance-imaging-based predictive models (MRI-PMs) improve the MRI prediction of clinically significant prostate cancer (csPCa) in prostate biopsies. Risk calculators (RC) provide easy individual assessment of csPCa likelihood. MRI-PMs have been analysed in overall populations of men suspected to have PCa, but they have never been analysed according to the prostate imaging-report and data system (PI-RADS) categories. Therefore, the true clinical usefulness of MRI-PMs regarding the specific PI-RADS categories is unknown.

**Abstract:**

A new and externally validated MRI-PM for csPCa was developed in the metropolitan area of Barcelona, and a web-RC designed with the new option of selecting the csPCa probability threshold. The development cohort comprised 1486 men scheduled to undergo a 3-tesla multiparametric MRI (mpMRI) and guided and/or systematic biopsies in one academic institution of Barcelona. The external validation cohort comprised 946 men in whom the same diagnostic approach was carried out as in the development cohort, in two other academic institutions of the same metropolitan area. CsPCa was detected in 36.9% of men in the development cohort and 40.8% in the external validation cohort (*p* = 0.054). The area under the curve of mpMRI increased from 0.842 to 0.897 in the developed MRI-PM (*p* < 0.001), and from 0.743 to 0.858 in the external validation cohort (*p* < 0.001). A selected 15% threshold avoided 40.1% of prostate biopsies and missed 5.4% of the 36.9% csPCa detected in the development cohort. In men with PI-RADS <3, 4.3% would be biopsied and 32.3% of all existing 4.2% of csPCa would be detected. In men with PI-RADS 3, 62% of prostate biopsies would be avoided and 28% of all existing 12.4% of csPCa would be undetected. In men with PI-RADS 4, 4% of prostate biopsies would be avoided and 0.6% of all existing 43.1% of csPCa would be undetected. In men with PI-RADS 5, 0.6% of prostate biopsies would be avoided and none of the existing 42.0% of csPCa would be undetected. The Barcelona MRI-PM presented good performance on the overall population; however, its clinical usefulness varied regarding the PI-RADS category. The selection of csPCa probability thresholds in the designed RC may facilitate external validation and outperformance of MRI-PMs in specific PI-RADS categories.

## 1. Introduction

Prostate cancer (PCa) suspicion is established from prostate-specific antigen (PSA) serum elevation and/or abnormal digital rectal examination (DRE), while its diagnosis is confirmed with a prostate biopsy [1]. The classic approach based on systematic biopsies results in high rates of unnecessary biopsies and overdetection of insignificant PCa (iPCa) [2]. The detection of clinically significant prostate cancer (csPCa) has improved with the use of magnetic resonance imaging (MRI) and guided biopsies [3,4]. However, that approach can improve by emploting the proper selection of prostate biopsy candidates after MRI [5,6]. For this purpose, PSA density (PSAD) [7], MRI-based predictive models (MRI-PMs) [8], and modern markers [9] have been recommended. MRI-PMs share Prostate Imaging-Report and Data System (PI-RADS) scores and additional independent predictors as PSAD [8,9,10]. External validation is always required before implementing any predictive model in new population [11,12]. To date, at least fifteen MRI-PMs have been developed [11,12,13,14,15,16,17,18,19,20,21,22,23,24,25]. In 10 of them, MRI results were reported using the latest PI-RADS versions 2.0–2.1 [11,13,19,20,21,22,23,24,25]; 5 of them had any external validation [20,21,22,24,25]; and none had any associated web or smartphone risk calculator (RC). In addition, the performance of MRI-PMs regarding the PI-RADS categories has never been analysed.

Our main objective was to design a new web-RC for csPCa likelihood, derived from a developed MRI-PM in the metropolitan area of Barcelona, Spain. The RC calculator will provide the novel option of selecting the csPCa probability threshold to facilitate future external validations and outperformances in specific PI-RADS categories. The specific objectives were: i. to develop the Barcelona MRI-PM; ii. to externally validate the developed model in a representative cohort of the Barcelona metropolitan area; iii. to design a friendly and free available web-RC with the PCa probability threshold selection allowed; and iv. to analyse the MRI-PM performance regarding PI-RADS categories in the development and external validation cohorts.

## 2. Materials and Methods

### 2.1. Development Cohort

The development cohort was formed of 1987 men with suspected PCa who had serum PSA > 3 ng/mL and/or abnormal DRE and were referred to our early PCa detection program from the primary care system. All men were scheduled for pre-biopsy multipa-rametric MRI (mpMRI), and thereafter, underwent guided and/or systematic biopsies from 1 to 4 weeks later, between 1 January 2016 and 31 December 2019, in one academic institution (VHH) of the metropolitan area of Barcelona, Spain. Data were collected in a prospective database according to the standards of reporting for MRI-targeted biopsy studies (START) [26]. Written consent was provided by all participants, and the institutional review board approved the project (PR/AG-317/2017). Men excluded from the study were: those undergoing 5-alpha reductase inhibitors for symptomatic benign prostatic hyperplasia; those with a previous PCa diagnosis; those with previous findings of atypical small acinar proliferation or prostate-intraepithelial neoplasia with atypia; and those with an incomplete data set. Additionally, 183 men were also excluded because mpMRI was not carried out due to technical reasons (56 due to claustrophobia, 32 due to a heart pacemaker, and 95 due to any metal prosthesis). The final development cohort comprised 1486 men (Figure 1).

### 2.2. External Validation Cohort

The external validation cohort comprised 946 men with suspected PCa who were retrospectively selected in two other academic institutions (PSM and GTIPH) and were representative of the metropolitan area of Barcelona, Spain. The criteria of PCa suspicion, the period of recruitment, and the diagnostic approach were the same as in the development cohort.

### 2.3. MRI Technique and Evaluation

MRI scans were acquired using a 3-tesla scanner with a standard surface phased-array coil. Magneto Trio (Siemens Corp., Erlangen, Germany) equipment was used for the development cohort and Diamond Select Achieva 3.0-TX (Phillips Corp., Eindoven, The Netherlands) and Nova Dual (Phillips Corp., Eindoven, The Netherlands) were used for the external validation cohorts. The acquisition protocol included T2-weighted imaging (T2W), diffusion-weighted imaging (DWI) and dynamic contrast-enhanced (DCE) imaging, according to the guidelines of the European Society of Urogenital Radiology. Two expert radiologists in each institution analysed images using the PI-RADSv.2.0, and in cases with multiple PI-RADS category lesions, the highest PI-RADS was selected for the model [27]. Prostate volume was assessed using MRI in all three institutions.

### 2.4. Prostate Biopsy Procedure

All men underwent a 2- to 4-core transrectal ultrasound (TRUS), cognitive-fusion guided biopsies for all PI-RADS > 3 lesions, and a 12-core TRUS systematic biopsy. Men with PI-RADS < 3 underwent a 12-core TRUS systematic biopsy. All biopsies were performed by one experienced urologist in each institution using the BK Focus 400 ultrasound scanner (BK Medical Inc., Herlev, Denmark) for the development cohort, and the Siemens Acuson 150 (Siemens Inc., Erlangen, Germany) and the Sonolite Antares (Siemens Inc., Erlangen, Germany) for the external validation cohort.

### 2.5. Pathologic Analysis and csPCa Definition

Biopsy samples were sent separately to local pathology departments, where two expert uro-pathologists analysed them, assigning them an International Society of Uro-Pathology (ISUP) grade group when PCa was detected. csPCa was defined when ISUP grade > 2 was founded [28].

### 2.6. Development of MRI-PM

The independent ability to predict csPCa of PI-RADSv.2.0 (1–5), age (years), ethnicity (Caucasian vs. others), serum PSA level (ng/mL), prostate volume (mL), DRE (normal vs. abnormal), PCa family history (no vs. first-degree relatives), and biopsy type (initial vs. repeat) was explored. PSAD was not directly included as a predictor because of the need for previous calculation, and due to having an area under the curve for csPCa detection of 0.892 and 0.897 when serum PSA and prostate volume were the predictors (data not shown).

### 2.7. Endpoint Measurements for the Performance Analysis of MRIPM

CsPCa detection rates and avoidable prostate biopsy rates.

### 2.8. Statistical Analysis

Descriptive statistics for the development and external validation cohorts were analysed and compared with the Chi-square and Mann–Whitney U tests. Binary logistic stepwise regression analysis of csPCa candidate predictors was performed for the model development. Continuous variables were modelled as linear or nonlinear predictors using restricted cubic splines. Calibration of the predictive model was assessed in both cohorts. Discrimination power was determined using the receiver operating characteristic (ROC) curve and the area under the curve (AUC), and clinical utility was determined using the clinical utility curve (CUC), which explored the potential rates of missed csPCa detection and avoidable prostate biopsies. The net benefit of the mpMRI- and MRI-based predictive model over biopsying all men was evaluated with a decision curve analysis (DCA). AUCs and specificities for 90% sensitivity to csPCa of the MRI and MRI-based predictive model in the development and external validation cohorts were compared with the DeLong and Chi-square tests, respectively. After selecting the threshold with 95% sensitivity to csPCa, the overall performances of the MRI and MRI-based predictive model were compared, and a sub-analysis—after stratifying by PI-RADS < 3, 3, 4 and 5—was performed. Sensitivity, specificity, positive and negative predictive values, and accuracy of rates of avoidable biopsies and potentially undetected csPCa were analysed. For the external validation, transparent reporting of a multivariable prediction model for individual prognosis or diagnosis (TRIPOD) statements were followed. Statistical analyses were computed using R programming language v.4.0.3 (The R Foundation for Statistical Computing, Vienna, Austria) and SPSS v.24 (IBM, statistical package for the social sciences, San Francisco, CA, USA).

## 3. Results

### 3.1. Characteristics of Development and External Validation Cohorts

The characteristics of development and external validation cohorts are summarised in Table 1. We noted a lower significant age and higher serum PSA in the development cohort than in the external validation cohort (*p* < 0.001). We also noted higher abnormal DRE, PCa family history, and repeat prostate biopsy rates in the external validation cohort (*p* < 0.001). The prostate volume was similar in both cohorts (*p* = 0.559). The Caucasian race was predominant in both cohorts (*p* = 0.738). Different case-mixes of PI-RADS categories existed between both cohorts (*p* < 0.001). We noted no significant increase in csPCa in the external validation cohort (*p* = 0.054), and a significant increase in iPCa (*p* < 0.001). The overall detection rate of csPCa was 36.9% in the development cohort, and regarding PI-RADS categories, 4.1% of men had PI-RADS < 3, 15.3% of men had PI-RADS 3, 52.4% of men had PI-RADS 4, and 83.4% of men had PI-RADS 5. The overall detection rate of csPCa was 40.8% in the external validation cohort, and regarding PI-RADS categories, 10.9% of men had PI-RADS < 3, 20.4% of men had PI-RADS 3, 51.9% of men had PI-RADS 4, and 84.0% of men had PI-RADS 5.

### 3.2. MRI-Based Predictive Model Development and Performance

Logistic regression analysis showed age, serum PSA, DRE, prostate volume, PCa family history, type of biopsy, and PI-RADSv2.0 as independent predictors of csPCa (Table 2), and a forest plot ranking the odds ratios is presented in Appendix A.

The ROC curves of mpMRI and MRI-PM in the development cohort are presented in Figure 2A. MRI-PM exhibited a net benefit of mpMRI over biopsying all men within the threshold, with a csPCa probability of 2%. MpMRI also exhibited a net benefit over biopsying all men (Figure 2B).

The AUC of the mpMRI increased from 0.842 (95% CI: 0.822–0.861) to 0.902 (95% CI: 0.880–0.914) of MRI-PM in the development cohort (*p* = 0.011). The efficacies of the mpMRI-PM were higher than those of mpMRI (*p* < 0.001) at 85%, 90%, and 95% sensitivities. At 90% sensitivity, the specificity of mpMRI was 56.8% (95% CI: 53.6–60.0) and that of MRI-PM was 69.5% (95% CI: 66.4–72.4; *p* < 0.001) (Table 3 (A)).

The MRI-PM derived nomogram is presented in Figure 3, and its calibration curve is presented in Appendix A.

CUCs showing the rate of avoidable biopsies and the corresponding rate of potentially missed csPCa, regarding the continuous csPCa probability threshold, are presented in Figure 4.

Details of avoidable biopsies and the corresponding risk of missing csPCa detection, regarding the probability threshold for a development cohort of 1000 men with suspected PCa, are displayed with absolute values in Appendix A (A), and relative values in Appendix A (A). The 15% threshold was selected due to its 95% csPCa sensitivity, which corresponded with a 40% rate of avoidable biopsies. In PI-RADS 5, all csPCas would be correctly classified, and 0.5% of biopsies would be avoided. In PI-RADS 4, 0.6% of csPCas would be missed, and 4% of biopsies would be avoided. In PI-RADS 3, 28.2% of csPCa would be missed, and 61.9% of biopsies would be avoided. Finally, in PI-RADS < 3, an MRI-based predictive model would suggest biopsy in 4.3% of cases, and 33.3% of extremely infrequent csPCa in this scenario (4.2%) would be detected (Table 4 (A)). The discrimination ability of the MRI-based predictive model regarding PI-RADS categories is shown in Appendix A, and its net benefits over biopsying all men regarding PI-RADS are presented in Appendix A.

### 3.3. External Validation of MRI-PM and Its Performance

The calibration curve of the MRI-PM shows how the nomogram slightly underestimated csPCa occurrence in the external validation cohort. For instance, for a 20% csPCa probability provided by the model (X axis), the real incidence is approximately 25%; thus, the model underestimates real csPCa occurrence. The intercept (0.261) and slope (0.815) show this disagreement, which is probably due to a 4% higher csPCa incidence (40.8 vs. 36.9%) in the cohort (Appendix A).

The ROC curves of MRI-PM in the external validation cohort and development cohort are presented in Figure 5A and DCA, showing the net benefit of MRI-PM in both the external validation cohort and the development cohort (presented in Figure 5B, respectively). The AUC of mpMRI in the external validation cohort was 0.743 (95% CI: 0.711–0.776) compared to 0.842 (95% CI: 0.822–0.861) in the development cohort (*p* < 0.001). The AUC of MRI-PM in the external validation cohort was 0.858 (95% CI: 0.833–0.883) while it was 0.897 (95% CI: 0.880–0.914) in the development cohort (*p* = 0.009) (Table 3 (A) and (B)). At 90% sensitivity for csPCa, the specificity of MRI-PM decreased from 52.3% to 41.3% in the external validation cohort (*p* < 0.001) (Table 3 (B)).

The CUCs of MRI-PMs in the external validation cohort are presented in Figure 4, where those in the development cohort are also presented. Details of avoided prostate biopsies regarding the csPCa probability thresholds, and the corresponding risk of non-detected csPCa for an external validation cohort of 1000 men, are displayed with absolute and relative values in Appendix A. At the 15% threshold, MRI-PM avoided 39.9% of prostate biopsies and missed 11.5% of csPCas. In PI-RADS 5, 2.4% of prostate biopsies would be avoided and 0.9% of csPCa would be missed. In PI-RADS 4, 7.3% of prostate biopsies would be avoided and 18.6% of csPCa would be missed. In PI-RADS 3, 63.2% of prostate biopsies would be avoided and 14% of csPCas would be missed. Finally, in men with PI-RADS < 3, MRI-PM would suggest biopsying 6.5% of them, detecting 18.2% of existing csPCa (Table 4 (B)). The discrimination ability of MRI-PM regarding PI-RADS categories is shown in the ROC curves in Appendix A, and the net benefits over biopsying all men regarding PI-RADS categories are presented in the DCA in Appendix A.

### 3.4. Web-RC Design

The validated nomogram was implemented in a web-diagnostic tool using RStudio v.1.2.5001 (RStudio Team, 2015; RStudio: Integrated Development for RStudio, Inc., Boston, MA, USA; URL: http://www.rstudio.com/; accessed on 12 April 2020) and the shiny library. The designed RC incorporates the novel option of selecting the csPCa probability threshold, and it is freely available at https://mripcaprediction.shinyapps.io/MRIPCaPrediction/ (accessed on 23 November 2020).

## 4. Discussion

Early detection of csPCa can improve even after the spread of MRI and guided biopsies. The proper individualised selection of candidates for prostate biopsy is the current challenge [6]. The most recent MRI-PMs include the latest versions of PI-RADS as well as the prostate volume derived from MRI [11,19,20,24]. Our model was developed from PI-RADS v2.0, which was always reported from 3-tesla mpMRI. The clinical independent predictors finally included in the developed MRI-PM, without a limited range, were the age, serum PSA, and prostate volume. We aimed to design an RC covering real clinical practice in contrast with others that limit the range of these predictors [12]. Ethnicity was not part of our developed model, since it was not an independent predictor of csPCa, perhaps because only one race was prevalent in both the development and external validation cohorts, in contrast to others [16]. PSAD, which is the most powerful predictor of csPCa after PI-RADS, has undergone increased use following the spread of MRI, providing the most accurate measurement of prostate volume without the additional cost [5,6,10]. Furthermore, PSAD appears to be an ideal predictor of csPCa for MRI-PM sharing due to its dynamic behavior across PI-RADS categories [7]. PSAD can be directly incorporated into models as a ratio [11,16,17,19,23,24], or indirectly through the serum PSA and the prostate volume, which also are independent predictors of csPCa [12,13,15,18,21,25]. Due to the observation of similar odds ratios of the two ways of expressing PSAD, we used serum PSA and prostate volume to avoid any calculation before the introduction of data into the RC. PSAD has been compared with some MRI-PMs, showing good performances compared to those MRI-PMs with inadequate calibration in the external validation cohorts [29]. We also included in our MRI-PM the type of biopsy (initial or repeat), since it was also an independent predictor and it provided greater clinical applicability than MRI-PM developed exclusively in biopsy-naïve men [20,21] or in men with previous negative biopsies [13,19,23,24].

External validation of developed model is a key point. It was carried out in a cohort selected in two representative institutions of the same metropolitan area where the MRI-PM was developed, using the same criteria of suspected PCa and the same diagnostic approach to csPCa. Even so, a non-significant 4-percentage-point difference in csPCa incidence of 36.9% vs. 40.8% was observed. This was stressful for the developed model, although its ability to discriminate csPCa remained accurate in the external validation cohort only, with minimal underestimation. We selected the 15% probability threshold of csPCa from development cohort due to its 95% sensitivity and avoidance of 40% of prostate biopsies. The same threshold provided a chance of avoiding 39.9% of biopsies with an 11.5% lack of csPCa detection in the external validation cohort. The clinical utility and net benefit of developed MRI-PMs over biopsying all men with PI-RADS > 3 was low in both the development and external validation cohorts. The avoidable biopsies would be 4% in PI-RADS 4 and 0.5% in PI-RADS 5 in the development cohort, and 7.3% and 2.4%, respectively, in the external validation cohort. CsPCa would be undetected in 0.6% and 0%, and 1.9% and 0.9%, respectively, in the development and external validation cohorts. The performance and net benefit in men with PI-RADS < 3 was better. In men with PI-RADS 3, 61.9% and 62.3% of prostate biopsies would be avoided in the development and external validation cohorts, respectively, while 28.2% and 14% of the few existing csPCas would be undetected. Finally, in men with normal mpMRIs (PI-RADS < 3), 4.3% and 6.5% would be biopsied, respectively, in the developed and external validation cohorts, detecting 33.3% and 18.2% of existing csPCa, respectively. In these low and intermediate PI-RADS categories, the absolute number of undetected csPCa is low due to the limited number of existing csPCa that did not reach rates higher than 5–10% of all detected csPCa.

External validations in populations where predictive models are going to be used are essential, and accessible and friendly RC are needed to avoid the cumbersome and time-consuming use of nomograms [11,12]. A novelty of our designed RC is the option to select the csPCa probability threshold for the overall population or according to the PI-RADS categories. The threshold can be adapted to the overall csPCa incidence of external validation populations [30]. In addition, the threshold selection can also improve the usefulness of the model in specific PI-RADS categories. For example, if we find it unacceptable that 28% of existing csPCa in men with PI-RADS 3 are not detected, we can select the csPCa probability threshold of 7%, which results in 11% of csPCa missing; however, it will result in 26% avoided prostate biopsies instead of 62%. This is the first time that the behavior of any MRI-PM for csPCa has been analysed according to the PI-RADS categories. This analysis shows that the usefulness of MRI-PM is limited in men with PI-RADS >3, and especially in men with PI-RADS 5, in whom the biopsy must be always. High PI-RADS categories exhibit high rates of csPCa; therefore, loosing small rates of csPCa, in addition to their greater aggressiveness, is dangerous [27,31,32,33,34,35].

We developed a new MRI-PM for csPCa and externally validated it, initially in the same metropolitan area. An accessible and friendly RC was designed for its easy and widespread use. The novelty of selecting the csPCa probability threshold may be helpful in new external validations and to modulate the desired sensitivity to csPCa in each PI-RADS category. Rather than limitations, we believe that local reporting by experienced radiologists following PI-RADS v.2 [27]; local experienced pathologists reporting ISUP grades [28]; and local experienced urologist performing biopsies according to the recommended EAU scheme [1] are strengths, because they represent real-life early detection of csPCa. We believe a true limitation of our MRI-PM is the prediction of csPCa made in prostate biopsies, which do not represent the true pathology in whole prostate gland [36]. Our model cannot be implemented in populations where ethnicity predicts different risk of csPCa. The applicability of our developed predictive model in other contemporary populations is unknown and should be evaluated in future studies; these include non-academic populations or in referred populations for prostate biopsy centers with a different case-mix of PI-RADS and csPCa incidence. Performing guided biopsies using a cognitive-fusion technique does not seem to be a limitation, since all men in the development and validation cohorts underwent the same technique, and no difference with the software fusion-guided technique has been found [37]. Further validations in cohorts in which other biopsy schemes and fusion techniques are used are needed. Finally, genomics must report a deeper understanding of PCa aggressiveness, and will help to improve the current csPCa definition. Thus, in the near future, radiomics will be able to predict the newly redefined csPCa [38].

## 5. Conclusions

A new MRI-PM for csPCa was developed to improve and individualise the selection of candidates for prostate biopsy in the metropolitan area of Barcelona, Spain. An associated web-RC incorporates the option to select the csPCa probability threshold, which may improve further external validations and outperformances in specific PI-RADS categories. The developed MRI-PM was able to detect 95% of existing csPCa, avoiding 40% of prostate biopsies in our overall population. However, the analysis regarding PI-RADS categories shows that the developed MRI-PM outperforms in PI-RADS < 3.

## Figures and Tables

**Figure 1 cancers-14-01589-f001:**
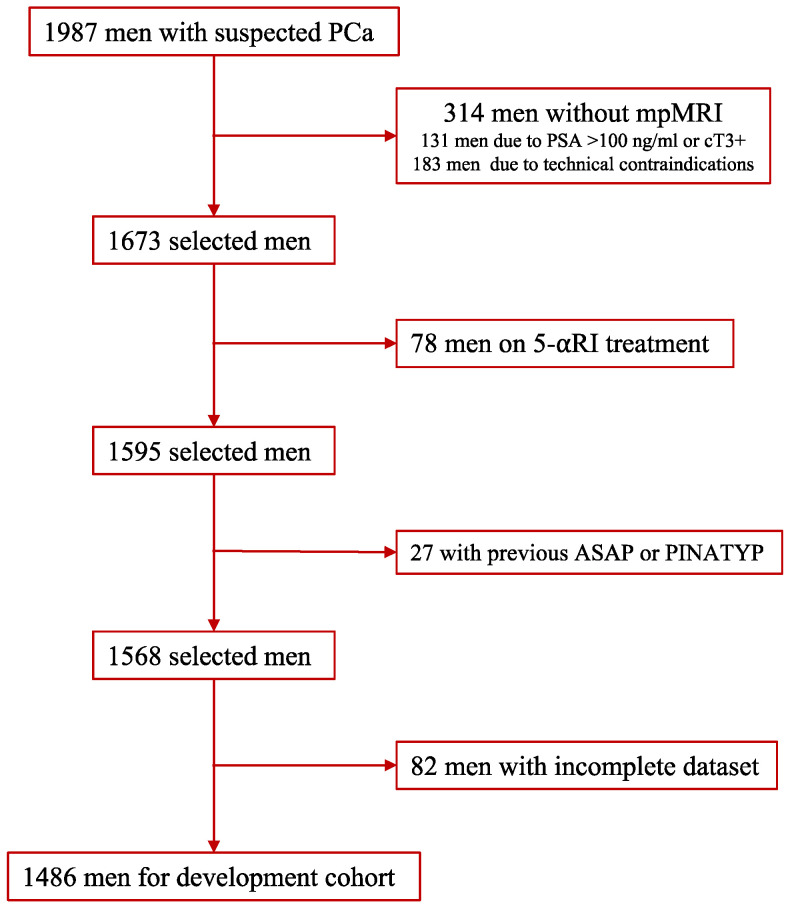
Flow chart of development cohort creation: inclusion and exclusion criteria.

**Figure 2 cancers-14-01589-f002:**
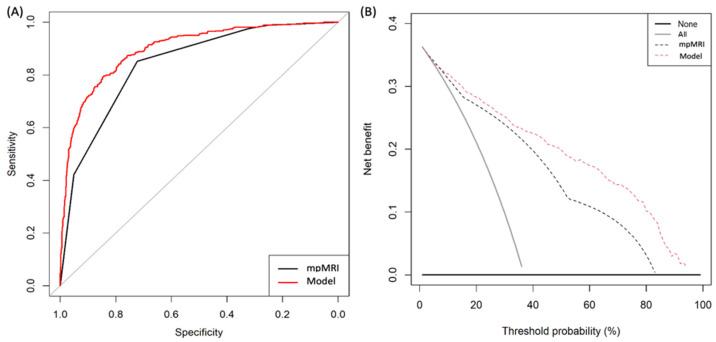
ROC curves showing the efficiency of MRI and MRI-PM in the development cohort (**A**). DCAs evaluating the net benefit of MRI and MRI-PM over biopsying all men belonging to the development cohort (**B**).

**Figure 3 cancers-14-01589-f003:**
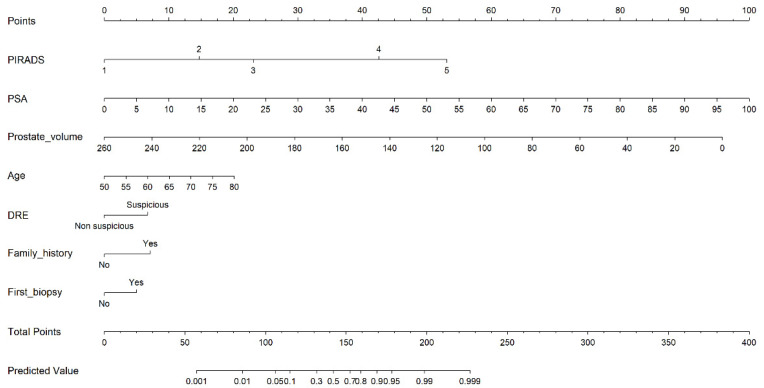
Nomogram derived from the developed MRI-PM model of csPCa in prostate biopsies.

**Figure 4 cancers-14-01589-f004:**
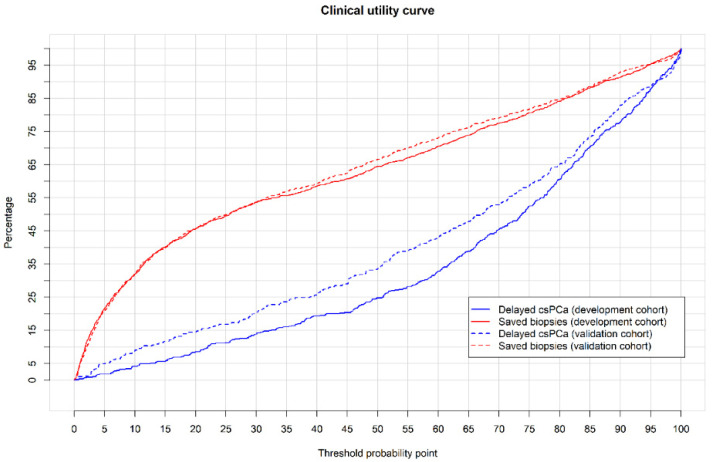
CUCs showing the rates of avoided biopsies (red lines) and corresponding missed csPCa (blue lines) regarding the continuous threshold of csPCa probability using MRI-PMs in development cohort (continuous lines) and external validation cohorts (interrupted lines).

**Figure 5 cancers-14-01589-f005:**
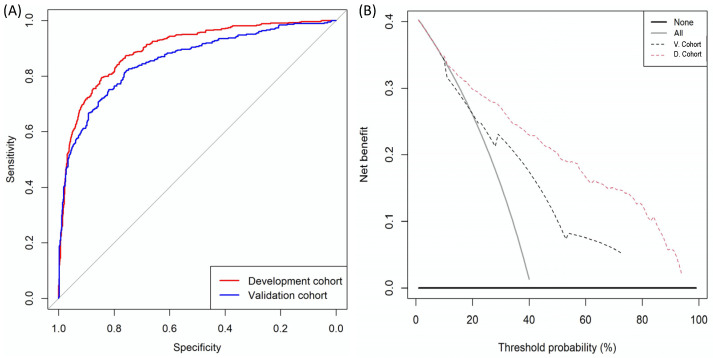
(**A**) ROC curves showing the efficacy of MRI-PM in development cohort and external validation cohort; (**B**) DCAs analysing the net benefit of MRI-PM in development (red interrupted line) and external validation (blue interrupted line) cohort over biopsying all men (continuous grey line).

**Table 1 cancers-14-01589-t001:** Characteristics of men suspected to have PCa in development and external validation cohorts and comparisons between them.

Characteristic	Development Cohort	External Validation Cohort	*p* Value
Number of men	1486	946	-
Caucasian race, *n* (%)	1465 (98.6)	931 (98.4)	0.738
Median age at biopsy (IQR), years	69 (62–74)	67 (61–72)	<0.001
Median serum PSA (IQR), ng/mL	6.0 (4.4–9.2)	7.4 (5.5–10.9)	<0.001
Abnormal DRE, *n* (%)	329 (22.1)	283 (29.9)	<0.001
PCa family history, *n* (%)	127 (8.5)	34 (3.6)	<0.001
Median prostate volume (IQR), mL	55 (40–76)	55 (40–78)	0.559
Prior negative prostate biopsy, *n* (%)	388 (26.1)	293 (31.0)	0.010
PI-RADS v.2.0, *n* (%)			
1	242 (16.3)	185 (19.6)	<0.001
2	73 (4.9)	50 (5.3)	
3	444 (29.9)	201 (21.2)	
4	450 (30.3)	391 (41.3)	
5	277 (18.6)	119 (12.6)	
PCa detection, *n* (%)	693 (46.6)	521 (55.1)	<0.001
csPCa detection, *n* (%)	548 (36.9)	386 (40.8)	0.054
iPCa detection, *n* (%)	145 (9.8)	135 (14.3)	<0.001

IQR = interquartile range; *n* = number; PSA = prostate-specific antigen; DRE = digital rectal examination; PI-RADS = Prostate Imaging-Reporting and Data System; PCa = prostate cancer; csPCa = clinically significant PCa; iPCa = insignificant PCa.

**Table 2 cancers-14-01589-t002:** Logistic regression analysis of independent significant predictors of csPCa in prostate biopsies.

Predictor	Odds Ratio (95% CI)	*p* Value
Age at prostate biopsy, ref. prior year	1.056 (1.036–1.077)	<0.001
Serum PSA, ref. prior ng/mL	1.085 (1.056–1.114)	<0.001
DRE, ref. normal.	1.730 (1.195–2.503)	0.004
Prostate volume, ref. prior mL	0.970 (0.964–0.977)	<0.001
Family history of PCa, ref. no	1.788 (1.066–3.002)	0.028
Biopsy type, ref. initial	0.668 (0.478–0.934)	0.018
PI-RADS v.2.0 score, 2 to ref. 1	3.311 (1.008–10.879)	0.048
3 to ref. 1	6.551 (2.740–15.661)	<0.001
4 to ref. 1	32.088 (13.660–75.377)	<0.001
5 to ref. 1	75.673 (30.738–186.311)	<0.001

CI = confidence interval; PSA = prostate-specific antigen; DRE = digital rectal examination; PI-RADSv.2 = Prostate Imaging-Reporting and Data System v.2.; ref. = referenced to.

**Table 3 cancers-14-01589-t003:** Efficacy of mpMRI and MRI-based predictive model analysed from the AUCs and specificities corresponding to the 85%, 90% and 95% sensitivity thresholds for csPCa, in development cohort (A) and external validation cohort (B).

Predictor	Development Cohort (A)				External Validation Cohort (B)			
	AUC (95% CI)	Specificities According to Sensitivity			AUC (95% CI)	Specificities According to Sensitivity		
		85%	90%	95%		85%	90%	95%
mpMRI	0.842 (0.822–0.861)	72.4 (69.4–75.2%)	56.8 (53.6–60.0)	40.7 (37.5–43.9)	0.743 (0.711–0.776)	45.5 (41.3–49.7)	41.3 (32.9–48.3)	14.3 (11.6–17.5)
MRI-PM	0.897 (0.880–0.914)	78.1% (75.3–80.7)	69.5 (66.4–72.4)	55.7 (52.5–58.9)	0.858 (0.833–0.883)	67.7 (63.6–71.5)	52.3 (48.1–56.5)	32.3 (28.5–36.4)
*p* Value	=0.011	*p* = 0.005	<0.001	<0.001	<0.001	<0.001	<0.001	<0.001

mpMRI = multiparametric magnetic resonance imaging; MRI-PM = MRI-based predictive model; AUC = area under the curve; CI = confidence interval.

**Table 4 cancers-14-01589-t004:** Clinical utility of MRI-based predictive model in terms of avoidable prostate biopsies and potentially missed csPCa in a 1000-sample-size development cohort (A) and external validation cohort (B), using the 15% threshold and regarding PI-RADS categories.

PI-RADS	Development Cohort (A)		External Validation Cohort (B)	
	Missed csPCa	Avoidable Biopsies	Missed csPCa	Avoidable Biopsies
1–2, *n* (%)	6/9 (66.7)	203/212 (95.7)	36/44 (81.8)	232/248 (93.5)
3, *n* (%)	13/46 (28.2)	185/299 (61.9)	6/43 (14.0)	134/212 (63.2)
4, *n* (%)	1/159 (0.6)	12/303 (4.0)	4/215 (1.9)	30/413 (7.3%)
5, *n* (%)	0/155 (0)	1/186 (0.5)	1/106 (0.9)	3/126 (2.4)
All, *n* (%)	20/369 (5.4)	401/1000 (40.1)	47/408 (11.5)	399/1000 (39.9)

## Data Availability

The data presented in this study are available on request from the corresponding author.

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
