# Peer review of "The Barcelona Predictive Model of Clinically Significant Prostate Cancer"

_cancers, 2022, doi:10.3390/cancers14061589_

Round 1

Reviewer 1 Report

The current study aims to design a new web-risk calculator of clinically significant prostate cancer (csPCa) derived from a MRI-based predictive model with clinical applicability in the metropolitan area of Barcelona, Spain.

The authors should be congratulated for the work and for addressing an important topic. However, few points warrant mentions:

Comments:

  1. Kim L et al. (PMID:32299423); (PMCID: PMC7164355); (DOI: 1186/s12916-020-01548-3) tested the use of the phi assay in standard clinical practice to pre-select men at the highest risk of harbouring significant cancer and hence refine the use of mpMRI and biopsies. They demonstrate that Phi as a triaging test may be an effective way to reduce mpMRI and biopsies without compromising detection of significant prostate cancers.

  1. The validation cohort is too small compared with the Development cohort. Authors should consider this.
  2. In line 62, “at 15 fifteen MRI-…”, there is probably a misspelling or it is not quite clear what you want to explain

  1. Since there aren’t for patients information about the time of diagnosis, this can affect the results. Authors should be clearer about the how long it takes from primary diagnosis to the first diagnostic imaging.
  2. It would appropriate to include the ISUP grade, repeatedly mentioned in the work and not fully described.

  1. I believe that authors should include more information also in view of current literature. This interesting paper deserve to read about:

   - https://pubmed.ncbi.nlm.nih.gov/34576134/ doi: 10.3390/ijms22189971.

         - https://pubmed.ncbi.nlm.nih.gov/33048224/; doi: DOI: 10.1007/s00261-020-02798-8

The authors should be congratulated for the work and for addressing an important topic. The manuscript is well written, and the methodology is robust and well described.

I believe that the study has sufficient merit to be considered for publication on Cancers, although major revisions are required.

Author Response

Reviewer 1- Comments and Suggestions for Authors

We sincerely appreciate all the comments and suggestions of the reviewer very much. Responses are below following them, in red color, as well as indicated the modifications that we have made in the manuscript, tables, figures, and references. All of modification carried out in the manuscript are in red.

Thank you very much

The current study aims to design a new web-risk calculator of clinically significant prostate cancer (csPCa) derived from a MRI-based predictive model with clinical applicability in the metropolitan area of Barcelona, Spain.

The authors should be congratulated for the work and for addressing an important topic. However, few points warrant mentions:

Comments:

  1. Kim L et al. (PMID:32299423);(PMCID: PMC7164355); (DOI: 1186/s12916-020-01548-3) tested the use of the phi assay in standard clinical practice to pre-select men at the highest risk of harboring significant cancer and hence refine the use of mpMRI and biopsies. They demonstrate that Phi as a triaging test may be an effective way to reduce mpMRI and biopsies without compromising detection of significant prostate cancers.

Effectively, as it is writing in the manuscript (line 59) modern markers as PHI are an option to improve the selection of candidates for prostate biopsy and, also to avoid MRI as we have observed also with Proclarix (Morote et al. PMID: 35021312; DOI: 10.5534/wjmh.210117. The use of new markers before MRI is interesting and it is supported in reference [9]. The inconveniences of using serum or urine determinations before MRI in men with suspected PCa findings as PCa markers are the cost and bothers men derived from additional determinations. Even more, MRI must be performed after positive results because the need of MRI in locating suspicious areas and performing guided biopsies. As we concluded in our article of Proclarix head-to-head analysis of different strategies are needed as well as cost-benefit analysis.

  1. The validation cohort is too small compared with the Development cohort. Authors should consider this. Thank for the comment. We believe that the only requirement for the size of development cohort is related with the multivariate models themselves. To our knowledge there is no criterion to establish the sample size for external validations. If we consider the criteria usually used for generation of multivariate models, Freeman suggested that number of cases should be greater than 10 * (k + 1), where k expresses the number of covariates. That is, the sample size had to be ten times the number of parameters/categories to be estimated plus one (Freeman DH. Applied categorical data analysis. New York: Marcel Dekker Inc; 1987.). And with a desirable balance of events at 50%. In our case, generation and external validation cohorts are large enough to follow these criteria. We agree with the reviewer that homogeneity between development and external validation cohorts is desirable, and the sample size of external validation cohort is conditioned by means of the parameters to validate, mainly the AUC. In our case, a sample size of 946 is great enough to provide a reasonable 95% confidence interval of AUC.

  1. In line 62, “at 15 fifteen MRI-…”, there is probably a misspelling, or it is not quite clear what you want to explain. Sorry for the mistake, it is corrected in the manuscript. “To date, at least fifteen MRI-based predictive models have been developed [11-25]”

  1. It would appropriate to include the ISUP grade, repeatedly mentioned in the work and not fully described. OK, we incorporate the term of “ISUP grade” instead of GG across the manuscript.

  1. I believe that authors should include more information also in view of current literature. This interesting paper deserve to read about:

- https://pubmed.ncbi.nlm.nih.gov/34576134/ doi: 10.3390/ijms2218997

- https://pubmed.ncbi.nlm.nih.gov/33048224/; doi: DOI: 10.1007/s00261-020-02798-8

Thank you for referencing these interesting articles, which have been introduced in the manuscript in the context of comment made in lines 323 [34], and in the new comment in lines 330-332 [36]

The authors should be congratulated for the work and for addressing an important topic. The manuscript is well written, and the methodology is robust and well described. I believe that the study has sufficient merit to be considered for publication on Cancers, although major revisions are required. Thank you.

Reviewer 2 Report

This study provides interesting new aspects regarding a possible use of an already externally validated MRI-based predictive model of csPCa, where first and repeat biopsy patients are included and where the probability threshold can be chosen.

Abstract:

I would highly recommend to correct this manuscript and especially the abstract with a native speaker. Mistakes from the Abstract:

Line 31 …and externally validate(d) in the…

Line 36 ...area under the cruve…(curve)  …

Line 38 …avoid 40.1% of (o)verall biopsies…

Lines 40-41…61.9% of biopsies would were avoided…

Line 43…overall performarce…

Introduction

Line 62 Authors have chosen uncommon or wrong wordings: …Until today, at 15 fifteen MRI- based predictive models have been developed [11-25]… Here they should also include the study by Deniffel D et al. Avoiding Unnecessary Biopsy: MRI-based Risk Models versus a PI-RADS and PSA Density Strategy for Clinically Significant Prostate Cancer, Radiology 2021 Aug;300(2):369-379.

Line 76 …development annd validation cohorts…

Material & Methods

Line 81 Double or missing spaces: …between   January 1of 2,016 and…

1) 183 men were excluded for technical reasons. What exactly does this mean?

Results:

In Table 1 and 2 the term prostate-specific antigen (not prostatic specific) should be used (see Introduction Line 51).

Line 154 …We noted higher significant age and serum PSA in external validation cohort…

I suggest to include: the external validation cohort. For the whole manuscript, again a native speaker should correct it.

Line 181 Table 3 …cohorts and Scheme 90. sensitivity… Please explain or correct this

2) When looking at Fig 2 A and Table 2 it is obvious that PIRADS 5 is the strongest parameter and that the model improves the PIRADS score from 2 to 4, within the sensitivity range of 40 to 90%. Several models provide specificities at 85, 90 and 95% sensitivity. Readers should be at least informed on specificity values at the high sensitivity 95%. I suggest two additional columns in Table 3.

3) The information for clinicians in Figure 4 is small. There are furthermore many abbreviations. Therefore, suggest, to include Figure 4 in the Supplement.

4) Why using numbers of a hypothetical cohort?

5) Why using a 15% threshold? Is it the Youden Index? Is it 90% sensitivity? Or is it the best relation between avoided biopsies and missed csPCa? Later in the discussion it is written: …The selected 15% probability threshold of csPCa in the overall development cohort exhibited 95% sensitivity, and it avoided 40% of prostate biopsies. … This information should be provided it the chapter results. Further, if 95% sensitivity is the probability threshold, I would highly suggest to include this in Table 3 (see Point 2).

6) Figure 6 is difficult to understand. The information might be omitted or placed in the supplement.

7) Line 215 Authors write: …The calibration curve of MRI-PM in the external validation cohort reflects the 4-percentage point increase of csPCa detection (40.8% vs. 36.9%), Figure 4(B)… I have difficulties to see this in Figure 4B, because the calibration line is above the ideal line only between 0.1 and 0.5 predicted probability. Which of the 14 numbers within this calibration curve reflects most this 4% increase? See Point 3.

8) Figure 2 and 7 both indicate the ROC curve from the development cohort. It is seen that the ROC curve from the validation cohort is in a similar range than the ROC curve from the PIRADS score. A true advantage for the model versus using PIRADS is therefore questionable. I would not only compare the development with the validation cohort but also the advantage versus PIRADS score at 90 and 95% sensitivity for both cohorts separately.

Discussion

9) The authors extensively discussed the possible role of PSA density in their model but provided no data at all. Authors have former publications on this topic. It would be of interest, if they can include PSA density into Table 2 to weight the value of this parameter despite PSA and prostate volume are included and independent values. This is interesting since the authors state: …The use of PSA density, which is the most powerful predictor of csPCa after PI-RADS… that PSA density is the 2nd best prediction parameter. It would be good, if the study by Deniffel D et al. Avoiding Unnecessary Biopsy: MRI-based Risk Models versus a PI-RADS… can be also discussed here, since they used PSA density in their model.

10) Authors write: …The clinical utility and net benefit of the developed MRI-based predictive model over biopsy of men with PI-RADS>3 was low in both development and validation cohorts which has never been reported. … For me this fact is clear because the ROC curve of the model in the validation cohort is similar to the ROC curve of the PIRADS score. Also, a true clinical usefulness is not very likely because usually all men with PIRADS 4 and 5 are highly recommended for prostate biopsy.

Line 285: …The performance and net benefit in men with PI-RADS<3 was better. …Here it should be written ≤3 because PIRADS 3 is included.

Line 290-1: …These rates of undetected csPCa are not few important because the rates of detected csPCa are not higher than 5%. … This sentence needs language correction.

11) It would be useful to know, in which real life cased authors use their own model now. I suggest to do this with PIRADS 3 lesions since PIRADS 4 and 5 lesions generally should be biopsied.

12) It is very positive that the parameter previous biopsy can be used since this allows a widespread use of this model. Further I congratulate for the significant novelty of their risk calculator for the option to select the csPCa probability threshold.

Finally, I checked the website on 09/03 https://mripcaprediction.shinyapps.io/MRIPCaPrediction/

It was good in overview, however, also here the authors should correct the mistakes in the sentences: This predictive model has demonstrated its external validity for diagnosing prostate cancer ISUP group ≥ 2(Gleason score ≥ 7).The approach for prostate biopsies was transrrectal: 

(missing space after 2) and transrrectal

Author Response

R2_Comments and Suggestions for Authors

We sincerely appreciate all the comments and suggestions of the reviewer very much. Responses are below following them, in red color, as well as indicated the modifications that we have made in the manuscript, tables, figures, and references. All of modification carried out in the manuscript are in red.

Thank you very much

This study provides interesting new aspects regarding a possible use of an already externally validated MRI-based predictive model of csPCa, where first and repeat biopsy patients are included and where the probability threshold can be chosen.

Abstract:

I would highly recommend to correct this manuscript and especially the abstract with a native speaker. Sorry for the inconveniences. Corrections have been done. Mistakes from the Abstract:

Line 31 …and externally validate(d) in the…Corrected in the manuscript

Line 36 ...area under the cruve…(curve)  … Corrected in the manuscript

Line 38 …avoid 40.1% of (o)verall biopsies… Corrected in the manuscript

Lines 40-41…61.9% of biopsies would were avoided… Corrected in the manuscript

Line 43…overall performarce… Corrected in the manuscript

Introduction

Line 62 Authors have chosen uncommon or wrong wordings: …Until today, at 15 fifteen MRI- based predictive models have been developed [11-25]… Corrected in the manuscript

Here they should also include the study by Deniffel D et al. Avoiding Unnecessary Biopsy: MRI-based Risk Models versus a PI-RADS and PSA Density Strategy for Clinically Significant Prostate Cancer, Radiology 2021 Aug;300(2):369-379. This article analyses the performance of some MRI-PMs and PSAD. It is very interesting, and we agree with the conclusion. The reference has been included the number [29]. The next comment has been included in lines 270-271: “PSAD has been compared with some MRI-PM showing good performances in front of not well calibrated MRI-PM [29]”

Line 76 …development and validation cohorts… Corrected in the manuscript

Material & Methods

Line 81 Double or missing spaces: …between   January 1of 2,016 and… Corrected i in the manuscript

183 men were excluded for technical reasons. What exactly does this mean?  It is referred to contraindications for MRI: (56) claustrophobia, (32) cardiac pacemaker, or (95) prosthesis. The next sentence has been added in lines 90-92: “Additionally, 183 men were excluded because MRI was not performed due to technical reasons (56 due to claustrophobia, 32 heart pacemaker, and prosthesis in 95)”.

Results:

In Table 1 and 2 the term prostate-specific antigen (not prostatic specific) should be used (see Introduction Line 51). Corrected in Tables 1 and 2

Line 154 …We noted higher significant age and serum PSA in external validation cohort…Sorry, it is a mistake. The sentence has been changed: “We noted lower significant age and higher serum PSA in development cohort”

I suggest to include: the external validation cohort. For the whole manuscript, again a native speaker should correct it. Done

Line 181 Table 3 …cohorts and Scheme 90. sensitivity… Please explain or correct this. Sorry, it is a mistake. The sentence is now: “Efficacy of mpMRI and MRI-based predictive model analysed from their AUCs and specificities corresponding to the 90% sensitivity thresholds for csPCa, in development cohort (A) and external validation cohort (B)”.

When looking at Fig 2 A and Table 2 it is obvious that PIRADS 5 is the strongest parameter and that the model improves the PIRADS score from 2 to 4, within the sensitivity range of 40 to 90%. Several models provide specificities at 85, 90 and 95% sensitivity. Readers should be at least informed on specificity values at the high sensitivity 95%. I suggest two additional columns in Table 3. Thank you for this suggestion. Specificities for 85% and 95% sensitivities have been in the new Table 3.

The information for clinicians in Figure 4 is small. There are furthermore many abbreviations. Therefore, suggest, to include Figure 4 in the Supplement. OK, Fig 4 is now included in the supplementary material as Figure S2

Why using numbers of a hypothetical cohort? This is referred to the estimation of results in 1000 size cohort which is usually used done with the objective of compare results. The term of hypothetical seems confuse and it has been suppressed in the manuscript and Tables S1 and S2

Why using a 15% threshold? Is it the Youden Index? Is it 90% sensitivity? Or is it the best relation between avoided biopsies and missed csPCa? Later in the discussion it is written: …The selected 15% probability threshold of csPCa in the overall development cohort exhibited 95% sensitivity, and it avoided 40% of prostate biopsies. …. The selection of the 15% threshold was based on its 95% csPCa sensitivity, and 40% was the corresponding avoidable biopsies rate in development cohort. Further, if 95% sensitivity is the probability threshold, I would highly suggest to include this in Table 3 (see Point 2). Yes, it has been included in the new Table 3. This information should be provided it the chapter results. Ok, it has been included in lines 199-201 with the next sentence: “The 15% threshold was selected due to its 95% csPCa sensitivity which corresponded with 40% rate of avoidable biopsies”

Figure 6 is difficult to understand. The information might be omitted or placed in the supplement. OK, Figure 6 is now placed as Figure S3. The figure presented the distribution probabilities of csPCa and benign tissue or iPCa in prostate biopsies of developed cohort (A) and external validation cohort (A), and their distribution according to the PI-RADS categories. It shows how the behavior of MRI-PM is different regarding PI-RADS, and how different thresholds should be selected for missing similar rates of csPCa in all PI-RADS categories.

Line 215 Authors write: …The calibration curve of MRI-PM in the external validation cohort reflects the 4-percentage point increase of csPCa detection (40.8% vs. 36.9%), Figure 4(B)… I have difficulties to see this in Figure 4B, because the calibration line is above the ideal line only between 0.1 and 0.5 predicted probability. Which of the 14 numbers within this calibration curve reflects most this 4% increase? See Point 3. Unfortunately, we have summarized excessively this result and the explanation must be complemented. We have introduced a new comment in lines 214-219, and we have modified the calibration curve figure to show only the two informative parameters: ‘intercept’ (calibration-in-the-large) that measure the difference between average predictions and average outcome; and ‘slope’; which reflecting the average effect of predictions on the outcome. The next sentence has been added in lines 217-222: “The calibration curve of the mpMRI-based PM shows how the nomogram underestimated slightly the csPCa occurrence in the validation cohort. For instance, for a 20% csPCa probability provided by the model (X axis), the real incidence is approximately 25%, thus the model underestimates real CsPCa occurrence. The intercept (0.261) and slope (0.815) show this disagreement, probably due to a 4% higher csPCa incidence (40.8 vs. 36.9%) in the cohort.”

Figure 2 and 7 both indicate the ROC curve from the development cohort. It is seen that the ROC curve from the validation cohort is in a similar range than the ROC curve from the PIRADS score. A true advantage for the model versus using PIRADS is therefore questionable. Thank you for the comment. This is not exactly what you say, because the AUC of mpMRI in external validation cohort was lower than that of development cohort. We have modified the sentence is in lines 214-218 to be best understanded: “The AUC of mpMRI in external validation was 0.743 (95% CI: 0.711-0.776) compared to 0.842 (95% CI: 0.822-0.861) in development cohort, p<0.001. The AUC of MRI-PM was 0.858 (95% CI: 0.833-0.883) in external validation cohort and 0.897 (95% CI: 0.880-0.914) of the development cohort (p=0.009)”. Effectively, in Fig.7 we present data of the external validation cohort compared to the development cohort. I would not only compare development with the validation cohort but also the advantage versus PIRADS score at 90 and 95% sensitivity for both cohorts separately. Ok, it is currently presented in Table 3.

Discussion

The authors extensively discussed the possible role of PSA density in their model but provided no data at all. Effectively, we discuss in lines 260-267 the usefulness of PSAD as it is an independent predictor of csPCa. We referenced that some predictive models have shared the calculated PSAD while others shared serum PSA and prostate volume separately as we did. The reason was the AUC of both possibilities were the same and introducing both variables in the risk calculator avoids its previous calculation. Additionally, PSAD has resulted the most powerful predictor of csPCa after PI-RADS score in all logistic regression models presented in the literature. We also confirmed this observation in our data when PSAD was directly included in the logistic regression (data nor reported). In Figure S1 we present the forest rank of the odd ratios of predictors in our MRI-model in Figure S1. Authors have former publications on this topic. It would be of interest, if they can include PSA density into Table 2 to weight the value of this parameter despite PSA and prostate volume are included and independent values This is interesting since the authors state: …The use of PSA density, which is the most powerful predictor of csPCa after PI-RADS… that PSA density is the 2nd best prediction parameter. Effectively, we are interested in PSAD, in fact we are now preparing a manuscript with a head-to-head comparison between PSAD and the MRI-predictive model regarding PI-RADS categories. We have not presented data of PSAD in the present manuscript. However, to include in the same logistic regression analysis PSAD, and PSA and PV seems not recommended due to co-linearity interferences. It would be good, if the study by Deniffel D et al. Avoiding Unnecessary Biopsy: MRI-based Risk Models versus a PI-RADS… can be also discussed here, since they used PSA density in their model. This is a very interesting study showing. in our opinion, two interesting facts must be remarked. One is the great utility of PSAD, and the other is the great difficulty of the external validations and calibrations of predictive models which are mainly produced by differences in csPCa incidences and differences on the weight of independent predictors between development and external validations cohorts. We have added this article in reference [29] and the next sentence have been added in lines 274-275: “PSAD has been compared with some MRI-PM showing good performances in front of not well calibrated MRI-PM [29]”.

Authors write: …The clinical utility and net benefit of the developed MRI-based predictive model over biopsy of men with PI-RADS>3 was low in both development and validation cohorts which has never been reported. … For me this fact is clear because the ROC curve of the model in the validation cohort is similar to the ROC curve of the PIRADS score. Also, a true clinical usefulness is not very likely because usually all men with PIRADS 4 and 5 are highly recommended for prostate biopsy. I agree with the reviewer comment. In fact, this is our conclusion. This is the first time that an MRI-PM has been analysed according to the PI-RADS categories. It is shown that efficacy and performance of the MRI-PM change regarding PI-RADS categories, we believe due to the different csPCa incidence.   

Line 285: …The performance and net benefit in men with PI-RADS<3 was better. …Here it should be writen ≤3 because PIRADS 3 is included. Thanks, it is corrected.

Line 290-1: …These rates of undetected csPCa are not few important because the rates of detected csPCa are not higher than 5%. … This sentence needs language correction. It has been corrected, the new sentence is: “In these PI-RADS categories, the absolute number of undetected csPCa was low because the rates were referred to an amount of csPCa not higher than 5% of overall csPCa detected”.

It would be useful to know, in which real life cased authors use their own model now. I suggest to do this with PIRADS 3 lesions since PIRADS 4 and 5 lesions generally should be biopsied. Effectively, we agree with the reviewer comment. This is our conclusion. We believe that this is a common fact for all MRI-PMs. However, as we state behind the behaviour MRI-PM regarding PI-RADS categories has not previously analysed.

It is very positive that the parameter previous biopsy can be used since this allows a widespread use of this model. Further I congratulate for the significant novelty of their risk calculator for the option to select the csPCa probability threshold. Thank you very much.

Finally, I checked the website on 09/03 https://mripcaprediction.shinyapps.io/MRIPCaPrediction/ It was good in overview, however, also here the authors should correct the mistakes in the sentences: This predictive model has demonstrated its external validity for diagnosing prostate cancer ISUP group ≥ 2(Gleason score ≥ 7).The approach for prostate biopsies was transrrectal: (missing space after 2) and transrectal. Thank you, corrections have been made.

Round 2

Reviewer 1 Report

The authors answered all the suggestions.